# Comparing a novel machine learning method to the Friedewald formula and Martin-Hopkins equation for low-density lipoprotein estimation

**Gurpreet Singh**[1¤a], **Yasin Hussain**[2], **Zhuoran Xu**[1], **Evan Sholle**[3], **Kelly Michalak**[1], **Kristina Dolan**[1], **Benjamin C. Lee**[1], **Alexander R. van Rosendael**[4], **Zahra Fatima**[1], **Jessica M. Peña**[1], **Peter W. F. Wilson**[5], **Antonio M. Gotto, Jr.**[6], **Leslee J. Shaw**[1], **Lohendran Baskaran**[1,7], **Subhi J. Al'Aref**[1¤b]*

1 Dalio Institute of Cardiovascular Imaging, Weill Cornell Medicine, New York, New York, United States of America, 2 Department of Internal Medicine, Yale University School of Medicine, New Haven, Connecticut, United States of America, 3 Information Technologies & Services Department, Weill Cornell Medicine, New York, New York, United States of America, 4 Department of Cardiology, Leiden University Medical Center, Leiden, The Netherlands, 5 Emory Clinical Cardiovascular Research Institute, Emory University, Atlanta, Georgia, United States of America, 6 Weill Cornell Medicine, New York, New York, United States of America, 7 Department of Cardiovascular Medicine, National Heart Centre, Singapore, Singapore

¤a Current address: Global Vx Tech, GlaxoSmithKline, Philadelphia, PA, United States of America
¤b Current address: Division of Cardiology, Department of Medicine, University of Arkansas for Medical Sciences, Little Rock, Arkansas, United States of America
* sjalaref@uams.edu

## Abstract

### Background

Low-density lipoprotein cholesterol (LDL-C) is a target for cardiovascular prevention. Contemporary equations for LDL-C estimation have limited accuracy in certain scenarios (high triglycerides [TG], very low LDL-C).

### Objectives

We derived a novel method for LDL-C estimation from the standard lipid profile using a machine learning (ML) approach utilizing random forests (the Weill Cornell model). We compared its correlation to direct LDL-C with the Friedewald and Martin-Hopkins equations for LDL-C estimation.

### Methods

The study cohort comprised a convenience sample of standard lipid profile measurements (with the directly measured components of total cholesterol [TC], high-density lipoprotein cholesterol [HDL-C], and TG) as well as chemical-based direct LDL-C performed on the same day at the New York-Presbyterian Hospital/Weill Cornell Medicine (NYP-WCM). Subsequently, an ML algorithm was used to construct a model for LDL-C estimation. Results are reported on the held-out test set, with correlation coefficients and absolute residuals used to assess model performance.

**Data Availability Statement:** All relevant data are within the manuscript and its Supporting Information files.

**Funding:** The research reported in this manuscript was supported by the Dalio Institute of Cardiovascular Imaging (New York, NY, USA). No funding was provided for this study. The funders had no role in study design, data collection and analysis, decision to publish, or preparation of the manuscript.

**Competing interests:** Gurpreet Singh became affiliated with GlaxoSmithKline after working on this project. Benjamin C. Lee receives consulting fees from Cleerly Inc, but has not receive that consulting fee since 2019. Leslee J. Shaw reports having an equity interest in Cleerly Inc. There are no patents, products in development or marketed products associated with this research to declare This does not alter our adherence to PLOS ONE policies on sharing data and materials.

## Results

Between 2005 and 2019, there were 17,500 lipid profiles performed on 10,936 unique individuals (4,456 females; 40.8%) aged 1 to 103. Correlation coefficients between estimated and measured LDL-C values were 0.982 for the Weill Cornell model, compared to 0.950 for Friedewald and 0.962 for the Martin-Hopkins method. The Weill Cornell model was consistently better across subgroups stratified by LDL-C and TG values, including TG >500 and LDL-C <70.

## Conclusions

An ML model was found to have a better correlation with direct LDL-C than either the Friedewald formula or Martin-Hopkins equation, including in the setting of elevated TG and very low LDL-C.

## Introduction

Atherosclerotic cardiovascular disease (ASCVD) is the leading cause of worldwide morbidity and mortality [1]. In the United States, annual mortality from ASCVD exceeds 800,000 deaths, while greater than 700,000 new cerebrovascular events occur annually, with an estimated cost of $351 billion [2]. Elevated low-density lipoprotein cholesterol (LDL-C) has been extensively validated as a major risk factor for the development of ASCVD [1]. Reduction in LDL-C has been shown to improve outcomes both within primary and secondary prevention cohorts [3, 4]. Multiple national and international clinical practice and societal guidelines, such as the American Heart Association/American College of Cardiology (AHA/ACC), European Society of Cardiology (ESC) and the Canadian Cardiovascular Society (CCS) consider LDL-C lowering as a primary target for both primary and secondary prevention [5–7]. In addition, contemporary data from novel lipid-lowering drug therapies show improved outcomes with aggressive LDL-C lowering beyond the traditional thresholds advocated for by current guidelines [8–11]. More recently, there has been a growing emphasis on residual cardiovascular risk in the setting of adequately controlled LDL-C levels, especially in the setting of elevated triglycerides [12]. As such, the clinical implications of LDL-C necessitate aiming for the most accurate estimates.

Traditionally, LDL-C has been estimated using the Friedewald formula, developed in 1972 on a cohort of 448 patients [13]. The equation estimates LDL-C as (total cholesterol [TC]) – (high-density lipoprotein cholesterol [HDL-C]) – (triglycerides [TG] / 5) in mg/dL [13]. A factor of 5 for triglycerides: very low-density LDL (TG: VLDL-C) was used for ease of computation in an era prior to the currently accepted LDL-C thresholds (Grundy, 2004). The Lipid Research Clinics Prevalence Study provided evidence of significant variance in the TG: VLDL-C ratio amongst individuals [14]. The Friedewald formula is particularly inaccurate for patients with low LDL-C levels or high triglycerides [15, 16]. To overcome these inaccuracies, in 2013, Martin et al. provided the Martin-Hopkins method for LDL-C estimation. The equation is (TC)–(HDL-C)–(TG/adjustable factor), where the adjustable factor stands for the strata-specific median TG: VLDL-C ratios [17]. The Martin-Hopkins method has been validated in multiple national and international trials [18–20]. This novel method has helped recategorize patients who were previously undertreated and is currently the method used for LDL-C estimation at multiple clinical laboratories [21]. However, the Martin-Hopkins equation was developed based on traditional linear regression analysis, and although it outperforms the Friedewald formula, there remain inaccuracies, especially at lower LDL-C estimates [22].

The accepted reference method for lipoprotein fraction measurement is the beta-quantification (BQ) method, which is possible in a limited setting but not suitable for mass screening due to its cost and labor-intensive nature. Machine learning (ML) utilizes sophisticated mathematical representation for the construction of inferential and predictive models. The use of ML has been shown to improve modeling and outcomes prediction in multiple domains within cardiovascular medicine [23, 24]. In an effort to further improve LDL-C estimation in this era of precision medicine, we derived a novel method for LDL-C estimation from the standard lipid profile using an ML approach based on the random forests algorithm (the Weill Cornell model). We compared the correlation between the Weill Cornell model to measured direct LDL-C along with the Friedewald and Martin-Hopkins methods for LDL-C estimation.

## Methods

The study sought to develop and subsequently validate a novel approach for the estimation of the serum LDL-C using ML-based random forests applied to routine cholesterol measurements, then compare its performance to the Friedewald formula and the more contemporary Martin-Hopkins equation.

### Study population

The study cohort comprised of a convenience sample of consecutive standard lipid profile samples (with the directly measured components of TC, HDL-C, and TG) as well as corresponding directly measured LDL-C values, performed between August 31st, 2005 and January 31st, 2019 for clinical indications at the New York-Presbyterian Hospital/Weill Cornell Medicine (NYP-WCM) inpatient and outpatient units across New York City and its boroughs. Inclusion criteria included determination of directly measured components of a standard lipid profile (TC, TG, HDL-C) as well as directly measured LDL-C on the same day in order to avoid day-to-day variations in cholesterol particles while comparing the calculated LDL-C value to the direct LDL-C. Further, we excluded lipid profiles with missing values for TC, HDL-C or TG. Data were extracted from the electronic health record (EHR) system using the Architecture for Research Computing in Healthcare (ARCH) program, a suite of tools and services offered by the Research Informatics team within NYP-WCM's department of Information Technologies & Services [25]. Since this study did not include personally identifiable information (PII), it did not constitute human subjects research and was deemed exempt from Institutional Review Board (IRB) review.

### Laboratory testing

Direct serum LDL-C measurement was performed using the Siemens ADVIA Chemistry XPT systems (Tarrytown, NY) at the NYP-WCM clinical laboratory. The clinical laboratory at NYP-WCM is regulated under the New York State Department of Health and is accredited by the College of American Pathologists (CAP). The assay was calibrated every 14 days and quality control measures followed government regulations or accreditation requirements. The assay was linear, measuring values from 8.0–1,670.0 mg/dl (0.21–43.25 mmol/L) with intra-assay coefficients of 0.4–0.5%. The limit of blank (LoB) for the ADVIA Chemistry assay was 0.1 mg/dL while the limit of detection (LoD) was 8.0 mg/dL. Serum total cholesterol, triglycerides and HDL-C were also measured using the Siemens ADVIA Chemistry XPT systems (Tarrytown, NY). The reportable range for total cholesterol was 10–1,350 mg/dl (0.55–74.93 mmol/L), and the assay was calibrated every 60 days, with 3 levels of quality control material that were analyzed twice daily. For triglycerides, similar calibration and quality control

standards were used. The reportable range for triglycerides was 10–1,100 mg/dl (0.55–61.05 mmol/L). For HDL-C, the assay was calibrated every 30 days, quality control measures were run twice daily, and the reportable range of HDL-C was 5–230 mg/dL (0.28–12.77 mmol/L). The ADVIA chemistry system assays for total cholesterol, direct LDL-C, triglyceride, and HDL-C are traceable to the National Cholesterol Education Programs / Centers for Disease Control (NCEP/CDC) reference method via patient sample correlation. The Friedewald and Martin-Hopkins estimated LDL-C values were calculated using the established and published formulas [13, 17].

## Machine Learning (ML)

ML analysis was performed using the application programming interface (API) of Scikit-learn [26]. A random forest model was constructed in order to predict the LDL-C value based on the values of the measured TC, TG and HDL-C (available through a standard lipid profile sample). The directly measured LDL-C served as the ground truth label. Random forest is a commonly used form of supervised learning that is employed for both classification and regression tasks. Random forests utilize tree representation in order to solve a problem wherein each leaf node corresponds to a class label. This algorithm was employed due to its state-of-the-art accuracy; interpretability; and lastly, its high degree of internal optimization compared with a relatively modest computational cost.

Overall, the original dataset was randomly split into a training (80%) and a held-out test set (20%). The training set was further divided into a derivation cohort (80%) and an interval validation cohort (20%), with results reported on the test set in order to confirm the validity of the findings. The model's hyper-parameters (number of nodes and depth of each tree) were fine-tuned using a randomized search with 10-fold cross-validation. During cross-validation, the training data was divided into equally sized subsets with training occurring on all but one of the subsets while internal validation was performed on the remaining subset. This process was repeated iteratively; i.e. in the case of 10-fold cross-validation, this step was repeated 10 times. Finally, the correlation between direct LDL-C and estimated LDL-C using the developed model (the Weill Cornell model) was evaluated, and subsequently compared to that of the Friedewald formula and Martin-Hopkins equation.

## Statistical analysis

Patient-level baseline clinical characteristics were collected for the study cohort using ICD-10 codes (E78.0–4 for hyperlipidemia, I10- I16 for hypertension, and I25.1 or I25.7 for coronary artery disease). Frequencies and proportions were calculated for categorical variables and means with standard deviations were calculated for continuous variables. All clinical data were analyzed on an aggregate basis, and at no point were individual patient comorbidities extracted or viewed. Correlation coefficients were used to compare model performance in predicting LDL-C value for each method. Subgroup analysis was performed with LDL-C and TG levels stratified according to ranges specified by the 2018 ACC/AHA Cholesterol guideline document [27]. Absolute residuals between subgroups across the 3 methods (the Weill Cornell model, Friedewald formula and the Martin-Hopkins equation) and the directly measured LDL-C level were compared using a paired Student's $t$-test, which provides both a measure of magnitude difference as well as directionality of the difference between the method and the ground truth direct LDL-C. Finally, LDL-C subgroup reclassification using the Weill Cornell model and compared to Friedewald and Martin-Hopkins method was performed using two by two tables. A one-tailed $p$-value of less than 0.05 was considered significant. All statistical analysis was performed using R version 3.5.0.

## Results

Between August 31st, 2005 and January 31st, 2019, there were 17,500 standard lipid profile samples paired with same-day direct LDL-C measures, performed on 10,936 unique individuals (4,456 female subjects; 40.8%) ranging in age from 1 to 103 (Table 1). 34.1% percent of patients from whom samples were drawn had been diagnosed with hypertension, 38% had been diagnosed with hyperlipidemia, and 12.8% had been diagnosed with coronary artery disease. Within the extracted samples, the mean direct LDL-C was 95.5 mg/dL and the mean triglyceride level was 59 mg/dL (Table 1).

Across all LDL-C levels, the Weill Cornell model exhibited a better correlation with direct LDL-C compared to the Friedewald formula or the Martin-Hopkins equation. Specifically, the correlation coefficient between the estimated and measured LDL-C value was 0.982 for the Weill Cornell model, compared to 0.950 for Friedewald formula and 0.962 for the Martin-Hopkins equation (Fig 1). In subgroup analysis, the Weill Cornell model was consistently superior across subgroups stratified by LDL-C and TG values, including TG >500 and LDL-C <70 (Table 2). Importantly, the magnitude of improvement was highest in the LDL-C >190 mg/dL strata (mean difference of -9.18 mg/dL compared to Friedewald formula and -8.81 mg/dL compared to Martin-Hopkins equation), while the Weill Cornell model was better at very low LDL-C levels (mean difference of -3.82 mg/dL compared to Friedewald and -1.84 mg/dL compared to Martin-Hopkins). Further, the Weill Cornell model showed improved performance compared to the Friedewald and Martin-Hopkins equations across triglyceride subgroups, with the largest magnitude of improvement in the triglyceride range of >500 mg/dL (mean difference of -27.17 mg/dL compared to Friedewald and -4.44 mg/dL compared to Martin-Hopkins). Fig 2 shows the scatter plot of estimated LDL-C vs. direct LDL-C stratified by the LDL-C subgroup, while Fig 3 shows the scatter plot stratified by triglyceride values. The correlation coefficient for the Weill Cornell model was 0.933, compared to 0.882 for Friedewald and 0.876 for Martin-Hopkins, in the LDL-C range of >190 mg/dL, while the correlation coefficient was 0.998 for the Weill Cornell model, compared to 0.942 for Friedewald formula and 0.901 for Martin-Hopkins equation, in the triglyceride range of >500 mg/dL.

In terms of reclassification, the Weill Cornell model resulted in the improved reclassification of LDL-C values across guideline-determined LDL-C thresholds compared to the Friedewald formula and Martin-Hopkins equation (S1 Table). For instance, there were 18 instances in the validation cohort where the Weill Cornell model correctly predicted an LDL-C in the

**Table 1. Patient-level baseline characteristics of the study cohort.**

| Clinical Variable | Value |
|---|---|
| Age in years (mean ± standard deviation) | 57.5 ± 16.9 |
| Female (%) | 40.8 |
| Mean height (in centimeters) | 170.2 |
| Mean weight (in kilograms) | 80.2 |
| Hyperlipidemia (%) | 38.0 |
| Hypertension (%) | 34.1 |
| Coronary artery disease (%) | 12.8 |
| Lipid particle | Value in mg/dL (mean ± standard deviation) |
| Total cholesterol | 171.0 ± 62.2 |
| High-density lipoprotein cholesterol (HDL-C) | 60.5 ± 3.5 |
| Triglyceride | 59.0 ± 33.9 |
| Direct low-density lipoprotein cholesterol (LDL-C) | 95.5 ± 64.3 |

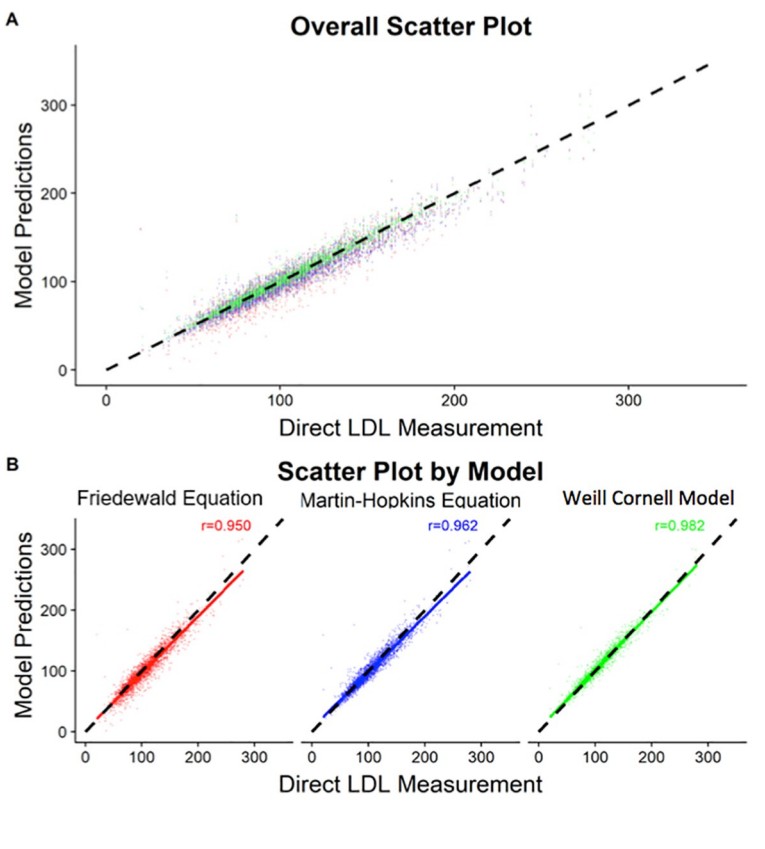

**Fig 1.** Scatter plot showing the correlation between estimated and directly measured low-density lipoprotein cholesterol (LDL-C) for the (a) overall cohort, and (b) for each of the LDL estimation models.

0–70 mg/dL range, while the Friedewald formula incorrectly predicted all 18 examples to be in the 70–100 range. Similarly, there were 15 cases where the Weill Cornell model correctly predicted an LDL-C in the 0–70 mg/dL range, while the Martin-Hopkins equation incorrectly predicted all 15 examples to be in the 70–100 range.

**Table 2. Comparison of the absolute residuals between estimated LDL-C using the Weill Cornell Model with the Friedewald formula and Martin-Hopkins equation.**

|  |  | Friedewald Formula |  | Martin-Hopkins Equation |  |
|---|---|---|---|---|---|
|  |  | Mean Difference (mg/dL) | *p* Value | Mean Difference (mg/dL) | *p* Value |
| **Overall** |  | -4.39±7.56 | <0.001 | -2.93±5.78 | <2.2e-16 |
| **LDL-C** | 0–70 | -3.82±8.15 | <0.001 | -1.84±6.35 | <0.001 |
|  | 70–100 | -3.72±6.51 | <0.001 | -2.22±4.21 | <0.001 |
|  | 100–130 | -4.12±7.43 | <0.001 | -2.67±5.30 | <0.001 |
|  | 130–160 | -5.02±7.86 | <0.001 | -3.90±6.62 | <0.001 |
|  | 160–190 | -7.49±9.24 | <0.001 | -6.19±7.50 | <0.001 |
|  | >190 | -9.18±9.77 | <0.001 | -8.81±9.38 | <0.001 |
| **TG** | 0–150 | -2.88±5.39 | <0.001 | -2.67±5.32 | <0.001 |
|  | 150–500 | -9.79±10.93 | <0.001 | -3.87±7.12 | <0.001 |
|  | >500 | -27.17±10.76 | 0.007 | -4.44±7.68 | 0.17 |

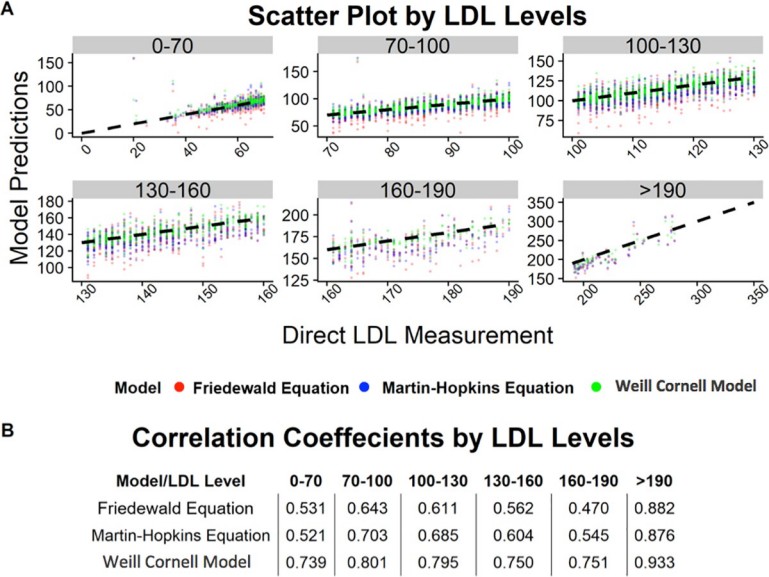

**Fig 2.** (A) Scatter plot showing the correlation between the ground truth LDL-C value (direct LDL-C) and estimated LDL-C value, across LDL-C subgroups, using the Weill Cornell model, Friedewald formula and Martin-Hopkins equation. (B) Correlation coefficients for each model for LDL-C subgroups.

## Discussion

LDL-C lowering has been a central target of primary and secondary prevention efforts. Furthermore, LDL-C lowering, and subsequent monitoring of LDL-C levels has become a key-stone of clinical practice, given the continuous and graded relationship between LDL-C levels and cardiovascular risk, as well as LDL-C lowering and subsequent modulation of incident

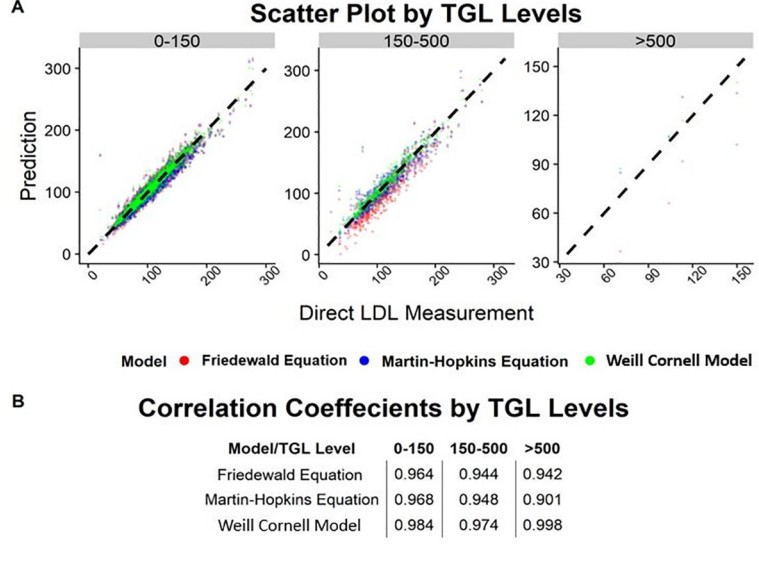

**Fig 3. Scatter plot showing the correlation between the ground truth LDL-C value (direct LDL) and estimated LDL-C value, across triglyceride subgroups, using the Cornell model, Friedewald formula and Martin-Hopkins method.** (B) Correlation coefficients for each model for TGL subgroups. Abbreviations: TGL: triglycerides.

risk. As a result, LDL-C measurement is ubiquitous within clinical care and accurate assessment is essential for the implementation of individualized treatment plans. In the present investigation, we used random forests in order to develop an ML-based approach for the estimation of serum LDL-C using standard lipid profile samples. We show that our model (the Weill Cornell model) had a better correlation with direct LDL-C when compared to the traditional Friedewald formula and the more contemporary Martin-Hopkins equation. Furthermore, our approach was consistently superior across subgroups stratified by LDL-C and triglyceride levels and resulted in a significant reclassification of LDL-C values across guideline-determined LDL-C thresholds compared to the Friedewald formula and Martin Hopkins equation. We developed and validated an approach that harnesses the power of ML-based algorithms for the estimation of serum LDL-C values. Further, estimated LDL-C using our ML-based model correlated better with direct LDL-C than both the Friedewald formula and Martin-Hopkins equations with very low LDL-C values (less than 70 mg/dL) or elevated triglyceride levels (> 500 mg/dL), which is significant in an era where lower LDL-C targets are sought after in highest-risk individuals using statins and novel non-statin drug therapies.

The BQ method, which combines ultracentrifugation with precipitation, is widely accepted as the reference method for lipoprotein fraction measurement. Yet, it is useful in limited settings and is not suitable for mass screening due to its cost and labor-intensive nature [28]. The enzymatic analysis of total cholesterol, triglyceride, and HDL-C is a considerably less costly procedure, and these measurements have been used to estimate the LDL-C using the Friedewald formula, thereby lowering overall costs and improving LDL-C integration within the clinical practice [13]. However, the Friedewald formula, from its inception, was known to be inaccurate in instances where triglyceride levels were greater than 400 mg/dL. In addition, major shortcomings of this approach include the requirement for a fasting specimen. Consequently, if non-fasting samples were used, there would be an overestimation of VLDL-C and underestimation of LDL-C (as a result of the presence of chylomicrons). In addition, due to its reliance on three combined measurements, LDL-C calculation is a product of their variabilities, with the largest effect being from total cholesterol measurements [29]. This variability ranges from 4% in well-standardized lipid laboratories to 12% in routine laboratories according to the National Cholesterol Education Program (NCEP) expert panel [30]. For instance, the LDL-C can often be estimated to be less than 70 mg/dL, despite directly measured levels being at greater than 70 mg/dL [31]. The Martin-Hopkins equation has largely replaced the Friedewald formula, using the same standard lipid measurements as the Friedewald formula but adding a personalized rather than a fixed conversion factor in calculating LDL-C [17]. The new formula is more reliable and can be used in non-fasting patients as it adjusts for triglyceride levels. The Martin-Hopkins method has been validated in multiple national and international trials and has helped re-categorize patients who would be undertreated using previous methods of LDL estimation [18–21]. The Martin-Hopkins equation is certainly more accurate than the Friedewald formula. However, even this improved equation is subject to inaccuracy at lower LDL-C estimates [22].

There has been widespread debate regarding optimal LDL-C targets. Despite the widespread use of statins, ASCVD remains the leading cause of worldwide mortality, while there remains residual cardiovascular risk despite optimal medical therapy [32]. The Pravastatin or Atorvastatin Evaluation and Infection Therapy–Thrombolysis in Myocardial Infarction 22 (PROVE IT-TIMI 22) trial noted residual cardiovascular risk despite lowering LDL-C to 62 mg/dL [33]. The hypothesis that more intensive treatment and lower LDL-C targets provide greater benefit has been further supported by plaque regression data from multiple studies [34–36]. A meta-analysis by the Cholesterol Treatment Trialists (CTT) showed that a 1 mmol/L reduction in LDL-C was associated with an approximately 20% reduction in major

cardiovascular events [8]. More recently, the IMPROVE-IT trial assessed the benefit of adding ezetimibe to statin therapy within a secondary prevention cohort, demonstrating that the addition of ezetimibe to statin therapy reduced mean LDL-C by 15.8 mg/dL (53.7 mg/dL in the combined therapy arm compared to 69.5 mg/dL in the statin monotherapy arm) with an associated absolute risk difference of 2% at 7 years, further supporting the "lower the better" argument for lower LDL-C targets [37]. On the other hand, there has been ongoing discussion regarding the long-term safety of lower LDL-C targets. In recent years, this debate has been further pushed into the spotlight after the approval for clinical use of monoclonal antibodies to proprotein convertase subtilisin/kexin type 9 (PCSK9), which achieve reductions of up to 60% in LDL-C levels and at times below 50 mg/dL [38, 39]. A recent meta-analysis of 3,340 patients on background maximally tolerated statin therapy and receiving a PCSK9 inhibitor showed that the incidence of treatment-related neurocognitive adverse event was low ($\leq$ 1.2%) with no significant differences between PCSK9 vs. control groups up to 104 weeks, with a similar finding in the subgroup of patients with LDL-C levels at <25 mg/dL [40]. Clinical practice will continue to evolve as evidence continues to accumulate regarding the beneficial effect of lower LDL-C targets. This, in turn, necessitates precise estimates of LDL-C to enable more accurate and well-informed clinical decision making, adverse event monitoring, and clinical trial design.

Machine learning can better generalize with the availability of larger datasets. In clinical cardiology, it has shown to be more proficient at predicting 5-year all-cause mortality than clinical characteristics or coronary computed tomography angiography (coronary CTA) metrics when used separately [41]. Machine learning has been used for segmentation tasks as well, with the goal of establishing the presence of a specific cardiovascular condition as well as a prognostication of outcomes on echocardiography, myocardial perfusion imaging, electrocardiography and coronary CTA [42–45]. It has been also used to answer complex and intricate clinical questions, such as the prediction of inpatient readmissions in heart failure patients [46]. In this analysis, we sought to further exploit the power of machine learning algorithms in order to answer a clinical question that has widespread implications for daily clinical practice. While typical models created using machine learning algorithms have an increasing number of variables, our approach was to simply utilize the standard lipid profile and its three measured components (TC, TG, and HDL-C) in order to estimate the serum LDL-C. Our approach and results provide further proof for the ability of machine learning algorithms to solve both common and complex clinical issues, especially when large volumes of data are available (central illustration).

## Central Illustration. Machine learning for the creation of an accurate model for serum LDL-C estimation

**Abbreviations:** TC: total cholesterol; TG: triglyceride; LDL: low-density lipoprotein; HDL: high-density lipoprotein.

This study was subject to some noteworthy limitations. Firstly, the Weill Cornell model was developed using a convenience sample of lipid profile measurements performed at a single-center tertiary care center in New York. While our model was internally validated, external validation is required in order to confirm the generalizability of the Weill Cornell model as well as its accuracy in other patient cohorts. However, the model is extremely portable, and it is reasonable to assume that adoption at other sites could yield similar performance, especially if a model were to be trained on a comparable data set. Secondly, the study cohort may not fully represent a general population since it is likely that there is a specific clinical indication for ordering a direct LDL-C and a standard lipid profile in the same setting. As such, selection bias cannot be excluded from this analysis. Thirdly, the present analysis focused on developing

and validating the Weill Cornell model across various LDL-C and TG levels but did not include patient-level analysis to determine the influence of certain clinical characteristics (such as ethnicity, presence of kidney disease, use of lipid-lowering drug therapies, etc.) on model performance. Nevertheless, ML has the ability to continuously update the algorithm as the model is applied to bigger and more diverse datasets, thereby creating a model that is unique in its accuracy, generalizability, and validity across all ethnicities and societies. Fourthly, direct LDL-C was determined using chemical-based methods, and not with the gold standard BQ method, while analysis was limited to correlation with direct LDL-C while true accuracy was not established. Nevertheless, the next step will be to validate the Weill Cornell model on cohorts with LDL-C measured by BQ. Finally, the model developed is not a simple score that can be calculated by a physician on the spot but requires computational processing. However, the existent paradigm involves the calculation of the estimated LDL-C when a standard lipid profile result is obtained after laboratory analysis, while the widespread digitization of health-care should obviate any obstacle to the implementation of our model. Integration of the Weill Cornell model into EHRs, many of which are already capable of implementing complex computational models for risk prediction and other tasks, could avert this limitation.

In summary, we developed the Weill Cornell model for LDL-C estimation using a random forest ML approach trained on measured components of a standard lipid profile (TC, HDL-C, and TG). Further, we observed that the Weill Cornell model correlates better with direct LDL-C than both the Friedewald formula and the Martin-Hopkins equation, with consistently better results across all subgroups, especially LDL-C <70 mg/dL and TG >500 mg/dL. Future research is required in order to validate the Weill Cornell model against LDL-C measured using the reference standard BQ method, with subsequent determination of model accuracy, beyond measures of correlation as shown in the present analysis. Such an approach is of critical importance in an era where accurate LDL-C is required as a result of more aggressive LDL-C lowering using novel and potent lipid-lowering drug therapies.

## Supporting information

**S1 Table.** Improved accuracy of LDL-C estimation using the Weill Cornell model results in significant reclassification of LDL-C values across guideline determined LDL-C thresholds (in mg/dL) compared to the (A) Friedewald formula and (B) Martin Hopkins equation. Results are shown for the validation set.
(DOCX)

**S2 Table.**
(XLSX)

## Author Contributions

**Conceptualization:** Gurpreet Singh, Yasin Hussain, Subhi J. Al'Aref.

**Data curation:** Gurpreet Singh, Zhuoran Xu, Evan Sholle.

**Formal analysis:** Gurpreet Singh, Subhi J. Al'Aref.

**Investigation:** Subhi J. Al'Aref.

**Methodology:** Gurpreet Singh, Subhi J. Al'Aref.

**Project administration:** Subhi J. Al'Aref.

**Resources:** Evan Sholle.

**Supervision:** Subhi J. Al'Aref.

**Validation:** Gurpreet Singh, Zhuoran Xu, Subhi J. Al'Aref.

**Visualization:** Zhuoran Xu, Subhi J. Al'Aref.

**Writing – original draft:** Subhi J. Al'Aref.

**Writing – review & editing:** Kelly Michalak, Kristina Dolan, Benjamin C. Lee, Alexander R. van Rosendael, Zahra Fatima, Jessica M. Peña, Peter W. F. Wilson, Antonio M. Gotto, Jr., Leslee J. Shaw, Lohendran Baskaran.

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
