## [Decision Letter · Decision Letter 0]

21 Jul 2020

PONE-D-20-07240

Comparing a novel Machine Learning method to the Friedewald formula and Martin-Hopkins equation for Low-density Lipoprotein Estimation

PLOS ONE

Dear Dr. Al'Aref,

Thank you for submitting your manuscript to PLOS ONE. After careful consideration, we feel that it has merit but does not fully meet PLOS ONE’s publication criteria as it currently stands. Therefore, we invite you to submit a revised version of the manuscript that addresses the points raised during the review process.

ACADEMIC EDITOR:

The authors should address the issues raised by reviewer 1 regarding additional computations (Kappa statistics and TeA) which would strengthen the value of the results.

We look forward to receiving your revised manuscript.

Kind regards,

Simeon-Pierre Choukem

Academic Editor

PLOS ONE

Journal Requirements:

2. Thank you for including your competing interests statement; "he authors have declared that no competing interests exist"

We note the following; 1) Author Benjamin C. Lee reports receiving consulting fees from Cleerly Inc. 2) Author Leslee J. Shaw reports having an equity interest in Cleerly Inc. and 3) Author Gurpreet Singh is affiliated to GlaxoSmithKline.

We note that one or more of the authors are employed by a commercial company: GlaxoSmithKline

Reviewers' comments:

Reviewer's Responses to Questions

**Comments to the Author**

1. Is the manuscript technically sound, and do the data support the conclusions?

Reviewer #1: Partly

Reviewer #2: Yes

2. Has the statistical analysis been performed appropriately and rigorously? 

Reviewer #1: No

Reviewer #2: Yes

3. Have the authors made all data underlying the findings in their manuscript fully available?

Reviewer #1: Yes

Reviewer #2: Yes

4. Is the manuscript presented in an intelligible fashion and written in standard English?

Reviewer #1: Yes

Reviewer #2: Yes

5. Review Comments to the Author

Reviewer #1: Permit me thank the authors for the huge effort put forth in a subject that is as important as it is relevant.

Summarily, this study seeks to derive a novel method that is more ACCURATE in estimating LDL-C using Machine Learning (ML).

1) The study portrays a sound scientific framework, but could be more so if a number of clarifications made:

As a limitation, the authors observed that, "direct LDL-C was determined using chemical-based methods, and not with the gold standard BQ method". This raise a question, What adjustment was done that allow the authors to compare a method (FF) that was established based on samples analyzed using the standard BQ method to those of the present study analyzed using a chemical-based method(that has been shown to have inherent inaccuracy when compared to the standard BQ)? https://doi.org/10.1177%2F107424840501000106. Is it possible that the novel formula could be reliable but not necessarily valid?

2) Statistics

If the interest is to derive a more ACCURATE method of LDL-C estimation, then it is a distraction to focus a lots of attention on describing the strong positive correlation between the novel and the reference method. Strong correlation does not necessarily mean accuracy.

secondly, applying Kappa statistics can help us answer the question on whether an observed agreement is by chance or not by chance.

Thirdly, if the observed agreement is not by chance, then assessing allowable total error (TEa) as a benchmark for performance of the novel method will be a great idea.

Reviewer #2: I have found the paper original as it attemps to resolve a key problem in the estimation of LDLc in people with cardiovascular diseases and other conditions with similar etiologies. The statistical analyses as well as all the steps of the procedure are appropriate and well described.

6. PLOS authors have the option to publish the peer review history of their article (what does this mean?). If published, this will include your full peer review and any attached files.

Reviewer #1: **Yes: **Tasha Manases

Reviewer #2: **Yes: **Pr Jules Clement Nguedia Assob

---

## [Author Response · Author response to Decision Letter 0]

26 Aug 2020

Reviewer Comments (Reviewer #1)

Permit me thank the authors for the huge effort put forth in a subject that is as important as it is relevant.

We would like to express our sincere appreciation to reviewer #1 for taking the time to review our manuscript and for providing comments that have helped us improve the manuscript.

Summarily, this study seeks to derive a novel method that is more ACCURATE in estimating LDL-C using Machine Learning (ML).

1) The study portrays a sound scientific framework but could be more so if a number of clarifications made: As a limitation, the authors observed that, "direct LDL-C was determined using chemical-based methods, and not with the gold standard BQ method". This raise a question, What adjustment was done that allow the authors to compare a method (FF) that was established based on samples analyzed using the standard BQ method to those of the present study analyzed using a chemical-based method(that has been shown to have inherent inaccuracy when compared to the standard BQ)? https://doi.org/10.1177%2F107424840501000106. Is it possible that the novel formula could be reliable but not necessarily valid?

We thank the reviewer for the excellent comment. We completely agree that an inherent limitation to our approach is that the “ground truth” LDL is not the accepted gold standard BQ method, which unfortunately has limited real life utility given the labor-intensive nature and significant associated cost. We sought to utilize real-world data for a proof-of-concept application in order to highlight the fact that machine learning can even improve upon relatively simplistic equations, but that applies to a relevant and daily aspect of patient care. We also wanted to highlight the fact that even though we developed an initial model with machine learning, a machine learning model can be dynamic and can become more accurate as more data is provided for model development. As such, future work is aimed at testing this model in an external validation cohort to further establish the advantage of using the methodology presented in this paper, as well as to use the data to further validate our model (by incorporating larger data, as well as data where the gold-standard LDL is measured using the BQ method). Additionally, the machine learning model used in this paper can be understood as multiple learners working together. On a fundamental level, this should allow the model to learn the underlying principle equation for calculating LDL levels and minimize the error procured by the model by learning a globally optimal equation.

To that end, we have highlighted this limitation in the discussion section:

1. Fourthly, direct LDL-C was determined using chemical-based methods, and not with the gold standard BQ method, while analysis was limited to correlation with direct LDL-C while true accuracy was not established. Nevertheless, the next step will be to validate the Weill Cornell model on cohorts with LDL-C measured by BQ.

2. Future research is required in order to validate the Weill Cornell model against LDL-C measured using the reference standard BQ method, with subsequent determination of model accuracy, beyond measures of correlation as shown in the present analysis.

2) If the interest is to derive a more ACCURATE method of LDL-C estimation, then it is a distraction to focus a lots of attention on describing the strong positive correlation between the novel and the reference method. Strong correlation does not necessarily mean accuracy.

We thank the reviewer for the comment. For continuous labels, mean absolute errors (MAE) is a good way to measure model accuracy. In this study, we compared the absolute prediction errors between the Weill Cornell model with Friedewald formula and Martin-Hopkins Equation by paired t test. We found that the Weill Cornell model has smaller absolute errors in both comparisons which implies higher accuracy. In addition, we utilized the matric of correlation coefficient to evaluate the model performance. It can represent the correlation extent on one hand. On the other hand, it equals to the square root of R square which can be interpreted as how much variance of the outcome can be explained by the predictor. So overall, we used two metrics to evaluate the model performance (in terms of accuracy as well as correlation).

3) Secondly, applying Kappa statistics can help us answer the question on whether an observed agreement is by chance or not by chance.

We thank the reviewer for the comment. Cohen’s kappa coefficient is a good measurement to evaluate inter-rater reliability, but it is typically used for categorical variables. In the present analysis, we treated LDL values as a continuous value (rather than splitting into categories since in real life the clinician is interested in the actual value rather than the range).

4) Thirdly, if the observed agreement is not by chance, then assessing allowable total error (TEa) as a benchmark for performance of the novel method will be a great idea.

We thank the reviewer for the comment. Congruent with the previous comment, we would like to clarify that our aim was to evaluate whether our model can predict an actual value rather than an actual class label.

 

Reviewer Comments (Reviewer #2)

I have found the paper original as it attempts to resolve a key problem in the estimation of LDLc in people with cardiovascular diseases and other conditions with similar etiologies. The statistical analyses as well as all the steps of the procedure are appropriate and well described.

We would like to express our sincere appreciation to reviewer #2 for taking the time to review our manuscript.

 

Again, we greatly appreciate the reviewer’s comments and hope that we have answered each point to his/her satisfaction.

Thank you very much for your consideration of this manuscript for publication. 

 Yours Sincerely,

 Subhi J. Al’Aref, MD, FACC (Corresponding Author)

---

## [Editor Report · Decision Letter 1]

16 Sep 2020

Comparing a novel Machine Learning method to the Friedewald formula and Martin-Hopkins equation for Low-density Lipoprotein Estimation

PONE-D-20-07240R1

Dear Dr. Al’Aref,

We’re pleased to inform you that your manuscript has been judged scientifically suitable for publication and will be formally accepted for publication once it meets all outstanding technical requirements.

Kind regards,

Simeon-Pierre Choukem

Academic Editor

PLOS ONE
---

## [Editor Report · Acceptance letter]

18 Sep 2020

PONE-D-20-07240R1 

Comparing a novel Machine Learning method to the Friedewald formula and Martin-Hopkins equation for Low-density Lipoprotein Estimation 

Dear Dr. Al’Aref:

I'm pleased to inform you that your manuscript has been deemed suitable for publication in PLOS ONE. Congratulations! Your manuscript is now with our production department. 

Kind regards, 

on behalf of

Dr. Simeon-Pierre Choukem 

Academic Editor

PLOS ONE